# A patient-oriented research approach to assessing patients' and primary care physicians' opinions on trauma-informed care

Seint Kokokyi[1]*, Bridget Klest[2], Hannah Anstey[2]

**1** Department of Clinical Health Psychology, Max Rady College of Medicine, University of Manitoba, Winnipeg, Manitoba, Canada, **2** Department of Psychology, University of Regina, Regina, Saskatchewan, Canada

* Seint.Kokokyi@umanitoba.ca

## Abstract

**Data Availability Statement:** All relevant data are within the manuscript and its Supporting Information files.

### Objective

To gather patients' and primary care physicians' (PCP) opinions on trauma-informed Care (TIC) and to investigate the acceptability of recommendations developed by patient, family, and physician advisors.

### Design

Cross-sectional research survey design and patient engagement.

### Setting

Canada, 2017 to 2019.

### Participants

English-speaking adults and licensed PCPs residing in Canada.

### Main outcome measures

Participants were given a series of questionnaires including a list of physician actions and a list of recommendations consistent with TIC.

### Results

Patients and PCPs viewed TIC as important. Both patients and PCPs rated the following recommendations as helpful and likely to positively impact patient care: physician training, online trauma resource centres, information pamphlets, the ability to extend appointment times, and clinical pathways for responding to trauma. PCPs' responses were significantly more positive than patients' responses.

**Funding:** B.K. 3134 Saskatchewan Health Research Foundation https://www.shrf.ca/ The funders had no role in study design, data collection and analysis, decision to publish, or preparation of the manuscript.

**Competing interests:** The authors have declared that no competing interests exist.

## Conclusion

TIC is important to patients and PCPs. Patients and PCPs believe changes to physician training, patient engagement, and systemic factors would be helpful and likely to positively impact patient care. Future research needs to be conducted to investigate whether these recommendations improve patient care.

## Introduction

Trauma-informed care (TIC) was developed based on research investigating service interventions for women survivors of violence with concurrent disorders [1] and has since been implemented in healthcare [2–5]. The Substance Abuse and Mental Health Administration (SAMHSA) described TIC as a framework that aims to understand and respond to the impact of trauma utilizing six key principles: 1) Safety, 2) Trustworthiness and Transparency, 3) Peer Support, 4) Collaboration and Mutuality, 5) Empowerment, Voice and Choice, and 6) Cultural, Historical, and Gender Issues [1]. There have been several reviews of TIC and its implications in the healthcare system [2–5]; however, there is variability in what is considered to constitute TIC in the provision of TIC training within the healthcare system, and in the literature discussing the need for TIC in healthcare practice. Further, there are concerns that TIC is simply good, patient-centered care and that implementing specialized training and practice would be redundant [3, 5, 6]. While some research has examined providers' knowledge and sense of efficacy regarding TIC [7, 8], there is limited investigation assessing both consumers' (i.e., patients) and providers' (i.e., physicians) opinions on TIC, whether they believe TIC is important, and whether or not TIC is already part of the care they receive or deliver. Lastly, patient engagement, which helps to tailor research to fit patient needs [9], has been inadequate in addressing this gap in previous studies of TIC.

The lifetime exposure to one or more traumatic event in Canada has been estimated to be 75.9% [10]. Trauma in this paper encapsulates definitions from two concepts: Criterion A of Posttraumatic Stress Disorder (PTSD) in the *Diagnostic Statistical Manual of Mental Disorders, 5th Edition* (DSM-5; actual or threatened death, serious injury, or sexual injury that is directly experienced, witnessed in person, happened to a family member or friend, or repeated exposure to averse details of these events [11]), and Adverse Childhood Experiences (ACEs; stressful and traumatic events that happened in an individual's first 18 years of life [12]). Trauma has been linked with psychological distress [13, 14], major depression, anxiety disorders, and PTSD [15–17]. Further, individuals who have experienced ACEs are at increased risk for multiple psychological (e.g., chronic depression, increased risk of suicide attempts, alcohol and substance use [12, 18] and medical illnesses (e.g., higher risk of somatic health disturbances, heart disease, liver disease, lung cancer, autoimmune disease [18]. Physicians are often the first line of response to these mental health problems, as individuals are more likely to seek mental health care from general physicians than from mental health professionals (e.g., counsellors, psychologists, psychiatrists [19]. The percentage of family physicians providing mental health care ranges from 40–53% depending on the population setting (urban or rural [20], and primary care physicians (PCPs) have identified that one-third of their caseload involves addressing patients' mental health concerns [21]. Given the unique position that PCPs are in when it comes to mental health care, assessing both PCP and patient opinions of TIC is of the utmost importance.

The current study was conducted with involvement from patient, family, and physician advisors. TIC was conceptualized using SAMHSA's framework and six core principles [1],

paraphrased by our team as Understanding Trauma, Safety (creating an environment that is physically and psychologically safe for everyone), Trust (building and maintaining trust with patients), Peer Support (connection to others for support), Empowerment (drawing on individuals' strength and experiences), Collaboration (leveling power between provider and patient), and Cultural Sensitivity (cultural responsiveness including awareness of different cultural beliefs and preferences, and addressing individuals' cultural needs). The study consisted of two phases. In Phase 1, the research team examined participants' opinions of each of the TIC core principles, specifically how important they view these principles and the perceived frequency with which they are practiced. These results informed a list of recommendations for TIC application for Phase 2. In Phase 2, patients' and physicians' evaluations of these recommendations were investigated, including their perceived usefulness.

## Phase 1

To our knowledge, no research has assessed the opinions of both healthcare consumers (i.e., patients) and providers (i.e., physicians) on TIC. Further, among the research that has investigated the implementation of TIC, the importance of TIC to consumers and providers and whether or not TIC is already being practiced has not been assessed. In an effort to narrow this gap, the opinions of patients and PCPs on TIC were explored in Phase 1. The general population (patient sample, henceforth referred to as patients) and PCPs (physician sample, herein referred to as physicians) in Canada were surveyed regarding their perspectives on TIC. Patients were asked how frequently they perceived receiving aspects of TIC and how important each of these aspects were to them. Similarly, physicians were asked how frequently they perceived delivering aspects of TIC and how important each of these aspects were to their practice. Phase 1 had three objectives: to investigate patient and physician opinions on the importance of aspects of TIC, to assess how often patients and physicians perceived receiving or delivering aspects of TIC, and to compare patient and physician responses.

## Methods

This study received ethics approval by the University of Regina's Research Ethics Board and consent was obtained in writing. Phase 1 used a cross-sectional research survey design to understand patient and physician opinions on TIC. Patients were 18 years of age or older, lived in Canada, and could read English. Recruitment was conducted through Qualtrics Panel, which allowed for a sample that was an approximate representation of the Canadian population with regard to gender, province, and ethnicity [22]. Patients completed a demographics (i.e., age, gender, ethnicity) questionnaire and the Patient TIC Survey. Any PCP practicing in Canada was eligible to participate. PCPs in Canada were recruited for the physician sample using purposive and snowball sampling methods, starting with known contacts. Approximately 850 physicians affiliated with medical colleges in Canada were also invited to participate in the study through email. Physicians were informed that a $20 donation (up to $1000) would be made to a patient advocacy organization for their participation in Phase 1. Physicians completed a set of demographic questions and the Physician TIC Survey.

In addition to the authors, the research team also consisted of two patient advisors, one family advisor, one PCP, and one volunteer research assistant who were actively involved throughout the research process. Each of the patient advisors had a personal history of trauma, and the family advisor had a family member with a trauma history. The physician advisor for this study attended medical school and completed residency training in Canada, and had been practicing as a family physician in Canada for several years.

The Patient and Physician TIC Surveys (S1 TIC Survey) comprise of 29 items that assesses the frequency and importance of TIC principles in primary care. These surveys were developed for the current study based on behavioural aspects of TIC principles. The questions were piloted on university students as well as physicians in the community, and were refined in collaboration with patient and physician advisors. Patients and physicians were asked to rate the frequency with which they received (patients) or provided (physicians) the indicated services and the importance of these services on two 5-point scales. Example questions include: "Make you feel welcome by being warm and friendly, and using a welcoming tone of voice"; and "Provide you with choices that fit your life circumstances for treatment preferences." Patients and physicians were also given an opportunity to provide written responses about trauma-informed patient care.

The TIC Survey average frequency and importance scores, were calculated for each subsample (patients and physicians), to yield the TIC Survey Average-Frequency and TIC Survey Average-Importance for each group. Subscale scores were calculated by averaging responses to sets of questions assessing each of the TIC principles as described in the introduction (henceforth referred to as aspects of TIC). Subscales for aspects of TIC were Understanding Trauma, Safety, Trust, Peer Support, Collaboration, Empowerment, and Cultural Sensitivity.

A repeated measures ANOVA was used to observe whether one aspect of TIC was favoured over another among patients and/or physicians. Additionally, within the physician subsample, if there were any discrepancies between importance ratings and frequency ratings, physicians were asked to check from a list of possible reasons to explain the disconnect.

## Results

Two hundred and ninety-six patients completed the study. The average age was 46.3 years, and 54.4% identified as female. Ethnicity and province of residence were proportional to the Canadian census. Sixty physicians completed the survey. Physicians' average age was 39.7 years, and the majority (61.4%) identified as female (36.8% male, and 1.8% "other"). Demographic information for patients and physicians can be found in Table 1.

A repeated measures ANOVA was used to compare the averages of the different TIC aspects among patients and physicians (Table 2). There were significant differences in how frequently patients perceived receiving TIC aspects and how important they viewed the same aspects. For both frequency and importance, the Trauma aspect was rated lowest by both patients and physicians. Significant differences were observed in which physician ratings were higher for how frequently they perceived delivering TIC aspects and how important they viewed these aspects compared to patients. Safety, Trust, and Collaboration aspects were rated higher than Trauma, Empowerment, Peer Support, and Cultural Sensitivity by both patients and physicians.

In regards to written responses, some patients commented on positive experiences with their PCPs, whom they described as caring and understanding. Others provided suggestions related to listening, communication, and having more time. One patient wrote, "Allow longer times for appointments, i.e.: do away with 10 minute limits. Treat whole patient, i.e.: do not limit problems to 2 per visit. Listen to the patient." Another commented, "More time to discuss issue/problem in detail or depth more time to discuss alternate treatment options or possibilities." Physician comments about general care include themes related to listening, time, and having empathy. For example, a physician said, "Time and resources is almost always a constraining factor." Another expressed, "Even if it means running late; I give these pts TIME an [sic] space to explore and definitely invite back for further discussion; I make sure to connect them to experts in the field."

**Table 1. Participants' demographic information.**

| Factor | M (SD) / Proportion* | | | |
|---|---|---|---|---|
| | Patients | | Physicians | |
| | Phase 1 | Phase 2 | Phase 1 | Phase 2 |
| N | 296 | 151 | 60 | 36 |
| Age | 46.28 (14.6) | 49.2 (13.1) | 39.70 (10.6) | 39.5 (10.3) |
| Gender | | | | |
| Female | 54.4% | 51.0% | 61.4% | 62.2% |
| Male | 44.9% | 48.3% | 36.8% | 37.8% |
| Other | 0% | 0% | 1.8% | 2.8% |
| Ethnicity | | | | |
| White | 43.6% | 45.0% | 66.7% | 63.9% |
| Aboriginal | 14.9% | 11.9% | 1.7% | 0% |
| Black | 4.4% | 3.3% | 0% | 0% |
| South Asian | 6.8% | 7.9% | 5.0% | 5.6% |
| East Asian | 17.9% | 23.2% | 10.0% | 11.1% |
| Southeast Asian | 1.4% | 0% | 5.0% | 5.6% |
| Filipino | 2.0% | 1.3% | 0% | 0% |
| Arab | 0.3% | 0.7% | 3.3% | 5.6% |
| West Asian | 1.0% | 0.7% | 0% | 0% |
| Latin American | 2.0% | 0.7% | 3.3% | 2.8 |
| Multiracial | 4.4% | 4.0% | 5.0% | 5.6 |
| Population Setting | | | | |
| Urban | 77.4% | 81.5% | 71.7% | 75.7% |
| Rural | 18.9% | 17.2% | 28.3% | 24.3% |
| First Nations Band | 3.4% | 1.3% | 1.7% | 0% |
| Province or Territory | | | | |
| British Columbia | 15.2% | 15.2% | 8.3% | 10.8% |
| Alberta | 9.1% | 7.9% | 45.0% | 45.9% |
| Saskatchewan | 10.5% | 10.6% | 20.0% | 21.6% |
| Manitoba | 4.7% | 4.0% | 0% | 0% |
| Ontario | 34.5% | 38.4% | 16.7% | 18.9% |
| Quebec | 7.4% | 7.3% | 3.3% | 0% |
| New Brunswick | 3.4% | 2.6% | 1.7% | 0% |
| Nova Scotia | 4.7% | 4.6% | 0% | 0% |
| Prince Edward Island | 3.0% | 2.6% | 0% | 0% |
| Newfoundland and Labrador | 4.7% | 4.6% | 0% | 0% |
| Yukon | 1.7% | 0% | 0% | 0% |
| Northwest Territories | 0.3% | 0.7% | 0% | 0% |
| Nunavut | 0.7% | 1.3% | 3.3% | % |

* Numbers may not add up to 100% as not all participants answered every question.

## Discussion

Overall, physicians reported delivering TIC at a higher rate than patients reported receiving it, and physicians viewed TIC as more important than patients did. Inquiring about trauma history was the lowest rated TIC aspect among both patients and physicians. It is possible that this is the case as patients and physicians may believe this type of screening is not relevant in a primary care setting. Relatedly, lack of awareness and training on how to screen for trauma

**Table 2. Repeated measures ANOVA for patients' and physicians' opinion on aspects of trauma-informed care.**

| Groups | Trauma | Safety | Trust | Peer Support | Collaboration | Empowerment | Cultural Sensitivity | Multivariate Test | Within-Subject Effects | Between-Subjects Effects |
|---|---|---|---|---|---|---|---|---|---|---|
| | $M$ (SD) | $M$ (SD) | $M$ (SD) | $M$ (SD) | $M$ (SD) | $M$ (SD) | $M$ (SD) | | | |
| **Frequency Ratings** | | | | | | | | | | |
| Patients | 2.53 | 3.92 | 4.01 | 3.48 | 3.90 | 3.67 | 3.20 | *TIC Factors:* | *TIC Factors:* | *Patient/Physician* |
| | (1.38) | (1.24) | (1.22) | (1.48) | (1.22) | (1.31) | (1.41) | Wilks' $\Lambda$ = .66 | $F(3.48,1190.71) = 63.92^{**}$ | $F(1,343) = 26.69^{**}$ |
| Physicians | 3.85 | 4.57 | 4.63 | 4.20 | 4.60 | 4.15 | 4.03 | $F(6,338) = 29.05^{**}$ | Partial $\eta^2$ = .16 | Partial $\eta^2$ = .07 |
| | (.76) | (.43) | (.37) | (.71) | (.31) | (.61) | (.70) | Partial $\eta^2$ = .34 | | |
| | | | | | | | | *Interaction:* | *Interaction:* | |
| | | | | | | | | Wilks' $\Lambda$ = .91 | $F(3.47, 1190.71) = 6.86^{**}$ | |
| | | | | | | | | $F(6,338) = 5.38^{**}$ | Partial $\eta^2$ = .02 | |
| | | | | | | | | Partial $\eta^2$ = .09 | | |
| **Importance Ratings** | | | | | | | | | | |
| Patients | 3.01 | 3.85 | 3.98 | 3.58 | 3.89 | 3.74 | 3.26 | *TIC Factors:* | *TIC Factors:* | *Patient/Physician* |
| | (1.34) | (1.13) | (1.13) | (1.29) | (1.12) | (1.19) | (1.35) | Wilks' $\Lambda$ = .81 | $F(3.44,1172.41) = 31.03^{**}$ | $F(1,341) = 43.65^{**}$ |
| Physicians | 4.37 | 4.69 | 4.73 | 4.47 | 4.72 | 4.49 | 4.33 | $F(6,336) = 13.59^{**}$ | Partial $\eta^2$ = .08 | Partial $\eta^2$ = .11 |
| | (.77) | (.42) | (.35) | (.78) | (.29) | (.61) | (.76) | Partial $\eta^2$ = .20 | | |
| | | | | | | | | *Interaction:* | *Interaction:* | |
| | | | | | | | | Wilks' $\Lambda$ = .95 | $F(3.44,1172.41) = 5.66^{**}$ | |
| | | | | | | | | $F(6,336) = 3.10$ | Partial $\eta^2$ = .02 | |
| | | | | | | | | Partial $\eta^2$ = .05 | | |

Note. Frequency scores: 1 = Never, 2 = Seldom, 3 = Occasionally, 4 = To a considerable degree, and 5 = Almost always. Importance scores: 1 = Not important,

2 = Slightly important, 3 = Moderately important, 4 = Important, and 5 = Very important.

$^{*}$ $p \leq .01$.

$^{**}$ $p \leq .001$.

and how to intervene may be a barrier for physicians [23, 24] and result in physicians viewing this aspect as less important. Given that trauma has been linked to numerous psychological and medical problems [18], screening for the impact of trauma instead of screening for trauma itself may be more beneficial [3], as physician knowledge of the impact of trauma can lead to appropriate referrals. Further, if physicians are able to apply TIC principles universally, simply knowing about patients' trauma history may not be helpful or necessary, as the insight into the impact of trauma garnered from applying TIC principles universally may be more relevant. Educating physicians on the impact of trauma may help them understand the significance of their patients' trauma history.

Trust, Safety, and Collaboration were the highest rated aspects of TIC among patients and physicians. Past research has demonstrated that trust in physicians and patient-physician collaboration are associated with greater patient satisfaction and positive patient health outcomes [25–27]. Without trust, patients may be less likely to share relevant information and may not follow through with treatment. Since TIC is a subset of patient-centred care, physicians may already be attuned to the importance of trust and collaboration in their practice. Physician

responses regarding safety may be related to the growing patient safety culture in primary care [28–30]. It is likely that physicians already have an interest in promoting a culture of safety and their responses reflected this interest.

## Phase 2

In Phase 1 both patients and physicians rated TIC as important for patient care, but patients did not perceive receiving TIC at the same level that physicians perceived administering it. The research team met to discuss the results from Phase 1 and to share their own experiences with the healthcare system, as consumers, providers, and family members. From these discussions, recommendations were made to address this discrepancy. The resulting recommendations were: physician training, booking longer appointment times, patient education, support groups for patients, and clinical pathways (i.e., guidelines for trauma assessment in primary care and treatment referral process [31, 32]. The aims of Phase 2 were to ask patients and physicians how helpful these recommendations would be and how likely they would be to use them if made available, and to compare patients' and physicians' responses.

### Methods

Approval for Phase 2 was obtained from the University of Regina's Research Ethics Board and consent was obtained in writing. A cross sectional survey research design was used in Phase 2 to investigate the acceptability of recommendations developed by the research team, surveying a subsample of patients and physicians who had previously participated in Phase 1.

The patient and physician surveys used in Phase 2 (S2 Phase 2 Recommendations) were developed based on the results from Phase 1 and discussions with the advisors. Recommendations included six types of physician training (i.e., TIC training, gender-based disparities in trauma and healthcare, racial disparities in trauma and healthcare, trauma and healthcare in marginalized groups, emotional skills, and self-compassion), billing mechanisms (i.e., to allow for booking ahead longer appointment times or to extend appointment times when situations arise unexpectedly), patient engagement (i.e., information pamphlets, peer support groups, and trauma resource centre), and clinical pathway with guidelines for assessment and treatment of trauma. Patients and physicians were asked how helpful they found each of the recommendations, the likelihood that the recommendations would positively impact patient care, and the likelihood that they would utilize these options if made available. For each of the recommendations on physician training, physicians were asked on a five-point Likert scale how likely they would be to attend the training if it were online, if in person, if continuing medical education (CME) credit was offered, and if CME credit was not offered. They were also asked how many CME credit hours would be ideal. A repeated measures ANOVA and *t*-tests were used to observe whether one recommendation was favoured over another among patients and physicians.

### Results

A total of 151 patients and 36 physicians from Phase 1 completed Phase 2. Participants' demographic information can be found in Table 1. Table 3 outlines mean patient and physician responses for physician training. There were no differences in physician and patient ratings across the physician training recommendations. Physician training in TIC was rated as more likely to positively impact care than physician training for gender-based disparities in trauma and healthcare. There were no differences between patients and physicians in how they responded to how helpful physician training in TIC would be or how likely this training would positively impact patient care. Physicians indicated they would be more likely to attend the

**Table 3. Repeated measures ANOVA for patients' and physicians' opinion on physician training.**

| Groups | TIC | Gender-Based Disparities | Racial Disparities | Marginalized Groups | Emotions | Self-Compassion | Multivariate Test | Within-Subject Effects | Between-Subjects Effects |
|---|---|---|---|---|---|---|---|---|---|
| | M (SD) | M (SD) | M (SD) | M (SD) | M (SD) | M (SD) | | | |
| **Helpfulness Ratings** | | | | | | | | | |
| Patients | 3.72 (1.06) | 3.76 (1.06) | 3.67 (1.14) | 3.65 (1.23) | 3.82 (1.16) | 3.81 (1.09) | *Training*: | *Physician Training*: | *Patient/Physician* |
| Physicians | 3.90 (.923) | 3.67 (1.06) | 4.13 (.90) | 3.97 (1.00) | 3.43 (1.28) | 3.70 (1.12) | Wilks' $\Lambda$ = .94 | $F(4.13,698.67) = 2.23$ | $F(1,169) = .11$ |
| | | | | | | | $F(5,165) = 1.95$ | Partial $\eta^2$ = .01 | Partial $\eta^2$ < .001 |
| | | | | | | | Partial $\eta^2$ = .06 | | |
| | | | | | | | *Interaction*: | *Interaction*: | |
| | | | | | | | Wilks' $\Lambda$ = .87 | $F(4.13,698.67) = 2.95^{**}$ | |
| | | | | | | | $F(5,165) = 4.88^{**}$ | Partial $\eta^2$ = .04 | |
| | | | | | | | Partial $\eta^2$ = .13 | | |
| **Likelihood Ratings** | | | | | | | | | |
| Patients | 3.71 (.92) | 3.45 (1.15) | 3.43 (1.13) | 3.45 (1.11) | 3.75 (1.02) | 3.74 (.98) | *Training*: | *Physician Training*: | *Patient/Physician* |
| Physicians | 4.03 (.93) | 3.60 (1.25) | 4.00 (1.17) | 3.77 (1.19) | 3.83 (1.12) | 3.67 (1.12) | Wilks' $\Lambda$ = .90 | $F(3.84,648.06) = 3.46^{*\dagger}$ | $F(1,169) = 1.65$ |
| | | | | | | | $F(5,165) = 3.69^{*}$ | Partial $\eta^2$ = .02 | Partial $\eta^2$ = .01 |
| | | | | | | | Partial $\eta^2$ = .10 | | |
| | | | | | | | *Interaction*: | *Interaction*: | |
| | | | | | | | Wilks' $\Lambda$ = .94 | $F(3.84,648.06) = 3.84$ | |
| | | | | | | | $F(5,165) = 2.26$ | Partial $\eta^2$ = .02 | |
| | | | | | | | Partial $\eta^2$ = .06 | | |

Note. Helpfulness scores: 1 = Not at all helpful, 2 = Slightly helpful, 3 = Moderately helpful, 4 = Very helpful, and 5 = Extremely helpful. Likelihood scores: 1 = Very unlikely, 2 = Unlikely, 3 = Neither likely nor unlikely, 4 = Likely, and 5 = Very likely.

$\dagger$ TIC training was rated as more likely to positively impact patient care than training for gender-based disparities ($p$ = .002).

$^{*}$ $p \leq$ .01.

$^{**}$ $p \leq$ .001.

training if CME credit was offered and the average ideal number of CME hours ranged between 2 and 3.

Results from a repeated measures ANOVA (Table 4) showed a significant within-subjects effect such that patients and physicians found extending appointment times to be more helpful and more likely to positively impact patient care than booking ahead longer appointments. There was between-subject effect such that physicians' ratings were significantly higher than patients' for all four ratings.

Regarding information pamphlets, the topics "How Trauma Impacts Physical and Mental Health" and "Understanding PTSD and PTSD Treatment" received the highest ratings among patients and physicians (Table 5). Patients' responses were significantly lower than physicians' for the helpfulness of, and the likelihood of referring to, these pamphlets.

With regards to peer support groups, there were no differences among patients in the likelihood of attending these groups in person or online for specific concerns, for general concerns, or for navigating the healthcare system (Table 6). For groups intended for trauma survivors

**Table 4. Repeated measures ANOVA results comparing patients' and physicians' responses for booking longer appointments ahead of time and extending appointment times unexpectedly.**

| Factor | Patients | | Physicians | | Multivariate Test | Within-Subject Effects | Between-Subjects Effects |
|---|---|---|---|---|---|---|---|
| | Booking Longer Appointments Ahead of Time | Extending Appointment Times Unexpectedly | Booking Longer Appointments Ahead of Time | Extending Appointment Times Unexpectedly | | | |
| | $M$ (SD) | $M$ (SD) | $M$ (SD) | $M$ (SD) | | | |
| Helpfulness | 3.59 (1.13) | 3.81 (1.06) | 3.04 (.97) | 4.36 (.76) | Wilks' $\Lambda$ = .935 | $F(1,185) =$ 12.86** | $F(1,185) =$ 14.14** |
| | | | | | $F(1,185) =$ 12.86** | Partial $\eta^2$ = .065 | Partial $\eta^2$ = .071 |
| | | | | | Partial $\eta^2$ = .065 | | |
| Likelihood | 3.70 (1.06) | 3.82 (1.05) | 4.36 (.87) | 4.58 (.65) | Wilks' $\Lambda$ = .926** | $F(1,185) =$ 14.75** | |
| | | | | | $F(1,185) =$ 14.75 | Partial $\eta^2$ = .074 | |
| | | | | | Partial $\eta^2$ = .074 | | |

Note. Helpfulness scores: 1 = Not at all helpful, 2 = Slightly helpful, 3 = Moderately helpful, 4 = Very helpful, and 5 = Extremely helpful. Likelihood scores: 1 = Very unlikely, 2 = Unlikely, 3 = Neither likely nor unlikely, 4 = Likely, and 5 = Very likely.

\* $p \le .01$.

\*\* $p \le .001$

with specific concerns, physicians were more likely to refer their patients to an in-person group than an online group. However, there were no differences in the likelihood of physicians referring their patients for an in person or an online group for general concerns or for navigating the healthcare system as a trauma survivor. Patients rated peer support groups for trauma survivors with specific concerns and for navigating the healthcare system as less helpful than physicians (Table 6). For all three peer support groups, patients' ratings of the likelihood of joining the groups in person or online were lower than physicians' ratings of the likelihood of referring their patients to these groups.

Patients responded that a trauma resource website would be more helpful than a hotline, and they indicated they would be more likely to use a website than a hotline (Table 7). There were no differences in how helpful they found each kind of resource or in how likely they thought these recommendations would positively impact patient care. Physicians' ratings were higher than patients' ratings for this recommendation.

Patients reported they would be less likely to refer to a clinical pathway for trauma than physicians. Results for clinical pathway can be found in Table 8.

## Discussion

Both patients and physicians reported that physician training in TIC would be helpful and would likely positively impact patient care. Patients may have these views given their position at the receiving end of healthcare and their first-hand experience with physicians' competence and compassion. Physicians may be aware of their level of competence as well as their areas for improvement and thus may view training as a way to advance their knowledge and skill set. Past studies [33–36] showed it is feasible to provide brief physician TIC training with favourable outcomes, such as increased patient-centeredness, knowledge about trauma and its impacts, and increased confidence in discussing trauma with patients. To facilitate the

**Table 5. Repeated measures ANOVA for patients' and physicians' opinion on information pamphlets.**

| Information Pamphlet Topic | Patients M (SD) | Physicians M (SD) | Multivariate Test | Within-Subject Effects | Between-Subjects Effects |
|---|---|---|---|---|---|
| **Helpfulness Ratings** | | | | | |
| Information on Different Kinds of Trauma and How to Cope with Trauma | 3.26 (1.30) | 3.85 (.78) | *Topics:* Wilks' $\Lambda$ = .80 | *Topics:* $F(6.72,1129.39) = 5.12^{**}$ | *Patient/Physician* $F(1,168) = 8.93^*$ |
| How Trauma Impacts Physical and Mental Health | 3.44 (1.30) | 4.15 (.74) | $F(9,160) = 4.51^{**}$ | Partial $\eta^2$ = .03 | Partial $\eta^2$ = .05 |
| Understanding PTSD and PTSD Treatment | 3.40 (1.32) | 4.06 (.74) | Partial $\eta^2$ = .20 | | |
| How Trauma Affects Relationships | 3.38 (1.35) | 3.94 (.78) | *Interaction:* | *Interaction:* | |
| Disasters and Traumatic Loss | 3.26 (1.37) | 3.59 (.96) | Wilks' $\Lambda$ = .87 | $F(6.72,1129.39) = 3.25^*$ | |
| Traumatic Stress and Substance Abuse Problems | 3.30 (1.37) | 4.03 (.67) | $F(9,160) = 2.78^*$ | Partial $\eta^2$ = .02 | |
| Intimate Partner Violence | 3.03 (1.44) | 4.18 (.67) | Partial $\eta^2$ = .14 | | |
| Trauma as a Result of Medical Errors | 3.10 (1.45) | 3.47 (1.05) | | | |
| When a Friend or Loved One has been Traumatized | 3.27 (1.36) | 3.68 (.94) | | | |
| Trauma Information for Parents | 3.19 (1.43) | 3.97 (.72) | | | |
| **Likelihood Ratings** | | | | | |
| Information on Different Kinds of Trauma and How to Cope with Trauma | 3.20 (1.30) | 4.00 (.66) | *Topics:* Wilks' $\Lambda$ = .78 | *Topics:* $F(6.75,1133.09) = 6.51^{**}$ | *Patient/Physician* $F(1,168) = 11.55^{**}$ |
| How Trauma Impacts Physical and Mental Health | 3.34 (1.27) | 4.18 (.64) | $F(9,160) = 5.04^{**}$ | Partial $\eta^2$ = .04 | Partial $\eta^2$ = .07 |
| Understanding PTSD and PTSD Treatment | 3.30 (1.35) | 4.09 (.58) | Partial $\eta^2$ = .22 | | |
| How Trauma Affects Relationships | 3.25 (1.40) | 3.97 (.68) | *Interaction:* | *Interaction:* | |
| Disasters and Traumatic Loss | 3.22 (1.33) | 3.58 (.94) | Wilks' $\Lambda$ = .85 | $F(6.75,1133.09) = 3.59^{**}$ | |
| Traumatic Stress and Substance Abuse Problems | 3.20 (1.38) | 4.09 (.68) | $F(5,160) = 3.06^*$ | Partial $\eta^2$ = .02 | |
| Intimate Partner Violence | 3.01 (1.38) | 4.15 (.71) | Partial $\eta^2$ = .15 | | |
| Trauma as a Result of Medical Errors | 3.01 (1.42) | 3.45 (1.06) | | | |
| When a Friend or Loved One has been Traumatized | 3.15 (1.35) | 3.73 (.98) | | | |
| Trauma Information for Parents | 3.08 (1.46) | 3.91 (.77) | | | |

Note. Helpfulness scores: 1 = Not at all helpful, 2 = Slightly helpful, 3 = Moderately helpful, 4 = Very helpful, and 5 = Extremely helpful. Likelihood scores: 1 = Very unlikely, 2 = Unlikely, 3 = Neither likely nor unlikely, 4 = Likely, and 5 = Very likely.

* $p \leq$ .01.

** $p \leq$ .001.

**Table 6. Descriptive statistics for patients' and physicians' responses to the helpfulness of peer support groups for trauma survivors and the likelihood of either joining or referring patients to in person and online groups.**

| Support Group Type | Helpfulness Score M (SD) | Likelihood Score | | t-test Scores comparing likelihood to attend groups in person or online |
|---|---|---|---|---|
| | | In Person Support Group M (SD) | Online Support Group M (SD) | |
| **Patients** | | | | |
| Specific Concerns | 3.19 (1.42) | 2.30 (1.33) | 2.49 (1.40) | $t(143) = -2.26$, $p = .025$, Cohen's $d = .148$ |
| General Concerns | 3.12 (1.38) | 2.41 (1.32) | 2.52 (1.37) | $t(143) = -1.60$, $p = .112$, Cohen's $d = .098$ |
| Navigating the Healthcare System | 3.07 (1.44) | 2.34 (1.29) | 2.48 (1.37) | $t(139) = -1.86$, $p = .065$, Cohen's $d = .118$ |
| **Physicians** | | | | |
| Specific Concerns | 4.35 (.77) | 4.42 (.75) | 3.79 (1.11) | $t(32) = 3.29$, $p = .002$, Cohen's $d = .671$ |
| General Concerns | 3.76 (1.08) | 3.82 (.98) | 3.58 (1.09) | $t(32) = 1.39$, $p = .174$, Cohen's $d = .234$ |
| Navigating the Healthcare System | 3.76 (.82) | 3.76 (.94) | 3.64 (.93) | $t(32) = .751$, $p = .458$, Cohen's $d = .130$ |

Note. Helpfulness scores: 1 = Not at all helpful, 2 = Slightly helpful, 3 = Moderately helpful, 4 = Very helpful, and 5 = Extremely helpful. Likelihood scores: 1 = Very unlikely, 2 = Unlikely, 3 = Neither likely nor unlikely, 4 = Likely, and 5 = Very likely.

provision of TIC, physicians should receive training in TIC, as well as training in gender-based disparities, racial disparities, marginalized groups, emotional intelligence, and self-compassion.

Given recent events highlighting racial disparities and health inequities [37], TIC has become highly relevant due to its focus on cultural, historical, and gender-based issues [38]. Understanding gender differences in healthcare including the power differential that exists between genders, differing levels of sensitivity to safety, and differing social roles, is integral to physicians creating spaces that are physically and emotionally safe for patients [39, 40].

**Table 7. Repeated measures ANOVA to investigate patients' and physicians' responses to hotline and website trauma resource centre.**

| Groups | Helpfulness | | | | Likelihood to use/refer | | | |
|---|---|---|---|---|---|---|---|---|
| | Hotline M (SD) | Website M (SD) | Multivariate test | Within-Subject Effects | Hotline M (SD) | Website M (SD) | Multivariate test | Within-Subject Effects |
| Patients | 3.43 (1.25) | 3.68 (1.11) | Wilks' $\Lambda = .869$ $F(1,144) = 21.77^{**}$ Partial $\eta^2 = .131$ | $F(1,144) = 21.77^{**}$ Partial $\eta^2 = .131$ | 2.95 (1.29) | 3.50 (1.21) | Wilks' $\Lambda = .748$ $F(1,144) = 48.45^{**}$ Partial $\eta^2 = .252$ | $F(1,144) = 48.45^{**}$ Partial $\eta^2 = .252$ |
| Physicians | 4.24 (1.01) | 4.35 (.88) | Wilks' $\Lambda = .788$ $F(1,33) = 8.87^{*}$ Partial $\eta^2 = .212$ | $F(1,33) = 8.87^{*}$ Partial $\eta^2 = .212$ | 4.38 (.85) | 4.56 (.82) | Wilks' $\Lambda = .933$ $F(1,33) = 2.36$ Partial $\eta^2 = .067$ | $F(1,33) = 2.36$ Partial $\eta^2 = .067$ |

Note. Helpfulness scores: 1 = Not at all helpful, 2 = Slightly helpful, 3 = Moderately helpful, 4 = Very helpful, and 5 = Extremely helpful. Likelihood scores: 1 = Very unlikely, 2 = Unlikely, 3 = Neither likely nor unlikely, 4 = Likely, and 5 = Very likely.

* $p \leq .01$.

** $p \leq .001$.

**Table 8. Independent *t*-tests comparing patients' and physicians' scores on clinical pathway and trauma resource centres.**

| Category | Recommendations | Helpfulness Scores | | | Likelihood Scores | | |
|---|---|---|---|---|---|---|---|
| | | Patients | Physicians | *t*-test scores | Patients | Physicians | *t*-test scores |
| | | *M (SD)* | *M (SD)* | | *M (SD)* | *M (SD)* | |
| Clinical Pathway | Clinical Pathway for Treating Trauma | 3.18 (1.27) | 3.79 (1.11) | *t*(177) = -2.53 | 3.28 (1.23) | 4.06 (.90) | *t*(178) = -3.44** |
| | | | | Cohen's *d* = .506 | | | Cohen's *d* = .725 |
| Resource Centre | Hotline | 3.44 (1.23) | 4.24 (1.02) | *t*(180) = -3.47** | 2.92 (1.30) | 4.38 (.85) | *t*(180) = -6.27** |
| | | | | Cohen's *d* = .700 | | | Cohen's *d* = 1.334 |
| | Website | 3.68 (1.11) | 4.35 (.88) | *t*(178) = -3.27** | 3.49 (1.22) | 4.56 (.82) | *t*(181) = -4.87** |
| | | | | Cohen's *d* = .665 | | | Cohen's *d* = 1.029 |

Note. Positive *t*-scores indicate higher patient ratings than physicians. Helpfulness scores: 1 = Not at all helpful, 2 = Slightly helpful, 3 = Moderately helpful, 4 = Very helpful, and 5 = Extremely helpful. Likelihood scores: 1 = Very unlikely, 2 = Unlikely, 3 = Neither likely nor unlikely, 4 = Likely, and 5 = Very likely.

* $p \leq$ .01.

** $p \leq$ .001.

Physician training in gender-based disparities related to trauma and healthcare could help physicians recognize any gender-related inequities within the healthcare system they operate in, leading to better patient care. Training in culturally-sensitive care may provide physicians with the tools they need to integrate patients' cultural meaning of health and illness in their practice. Further, training could help physicians consider how patients' experiences of racism and discrimination influence their health, access to healthcare, and quality of life [41]. Individuals belonging to multiple marginalized social categories experience more social obstacles than individuals who only belong to one category [42]. Further, there is a growing body of research [43–45] that demonstrates that trauma-informed care approaches may address the stress that comes with marginalization [44]. Subsequently, physician training on marginalized groups may allow physicians to understand the discriminatory experiences of patients with intersecting statuses of marginality and subsequently use this understanding to respond appropriately to patient needs.

Various other trainings related to TIC, such as ones that promote emotional intelligence and self-compassion, may be helpful for physicians in providing trauma-sensitive care. Physician training aimed at increasing emotional intelligence and skills could help physicians better understand, recognize, and respond to emotional expression in themselves and in their patients [46–48]. Physician compassion has been associated with patient trust [49] and increased collaboration with their patients [50]. Physicians who receive training in self-compassion have an increased awareness of how their presence impacts patients and are more likely to provide nurturing care to their patients [51]. Increasing physician compassion through self-compassion training could lead to better patient care, trust, and collaboration. This training is supported in a toolkit for trauma-informed primary care [52], which identified building compassion as a way to identify and to mitigate any physician burnout or compassion fatigue.

Physicians rated changes to billing mechanisms as more helpful and more likely to positively impact patient care than patients. Patients may not understand how physicians are compensated or the mechanisms behind billing, and resultingly, may not think recommendations about billing are as important to their care. For both samples, having a mechanism in place to extend appointment time due to unexpected situations was perceived as more helpful and more likely to positively impact patient care than booking a longer appointment time in advance. Some patients may need more time to have their emotional needs met, particularly if

triggered during a visit. As a result, some physicians may grant patients extra time at the cost of delaying remaining appointments, which may have a cumulative effect over the course of a day. While patients expect to wait for their appointments, the uncertainty of the wait can cause angst and stress, and make it seem longer [53]. Other physicians may not be able to provide patients with extra time and may end appointments sooner than what patients would like, causing patients to feel rushed and unsatisfied with the care they received. Shorter appointment times are associated with physician burnout and reduced feelings of productivity, particularly when managing complex multimorbid patients [54]. Thus, to address these difficulties, billing should focus on counseling to accommodate and encourage trauma-informed visits, which in turn could help physicians become more trauma-informed.

Physicians rated all patient engagement recommendations (i.e., information pamphlets, peer-support groups, and trauma resource centre), as more helpful and more likely to positively impact patient care than patients. Physicians may view the recommendations as both helpful and positive as they can think of numerous patients who might benefit from them. In contrast, each individual patient may only find certain recommendations to be personally relevant or beneficial and thus may rate the other recommendations as less helpful and impactful. The information pamphlets "How Trauma Impacts Physical and Mental Health" and "Understanding PTSD and PTSD Treatment" received high ratings for how helpful and how likely they would be referred to by both patients and physicians. "Understanding PTSD and PTSD Treatment" [55, 56] already exist, and information on "How trauma impacts physical and mental health" could be developed. Patients' ratings for the likelihood of referring to the other potential pamphlet topics were low, and thus they may not be as useful to have readily available.

While patients in this study reported they would find peer support groups moderately helpful, they indicated they were unlikely to attend any of the suggested groups, online or in person. There are many benefits to peer support groups, such as normalization, stigma reduction, getting support and being supported, promoting hope and belief in the possibility of a recovery, and promoting of help seeking behaviours and treatment engagement [57–59]. There are barriers as well, such as fear of harm to others or being harmed, feeling different from others in the group, fears of being judged, concerns with boundaries, concerns with misleading information, feeling uncertain about one's health, and facing hostile or derogatory comments from others [57–59]. There are also emotional costs to participating in peer support groups, such as retriggering past experiences and resurfacing of unhealed issues [60]. It is possible that patients in this study may believe the barriers to attending these groups outweigh the benefits. Thus, despite positive physician perceptions, peer support groups may not be useful for patients.

Patients indicated trauma resource websites would be more helpful and more likely to positively impact patient care than trauma resource hotlines. Research has shown that individuals use the Internet to seek health information [61–63]. Benefits to online resources include increased access to care [64] and reduced travel costs [65]. Online resources also offer confidentiality for individuals who have concerns about anonymity [66]. Physicians reported that online and over the phone trauma resource centres would be similarly helpful and likely to positively impact patient care. Given the positive reception to this recommendation, development and implementation of comprehensive online psychological support programs for trauma survivors in Canada should be considered.

Physicians in this study rated clinical pathways for trauma care as more helpful and more likely to positively impact care than patients. Patients may not have an understanding of clinical pathways and their impact on healthcare (e.g., help physicians with disease management and provide information on available services [31, 32] Patients may also not know what services exist. Physicians are able to provide choices to their patients when they know what

resources are available. As there is currently no research on the impact of clinical pathways on TIC, this is an area for future research consideration.

## General discussion

On average, patients and physicians in this study found TIC to be important. Patients reported receiving aspects of TIC service, but not nearly as frequently as physicians reported delivering them. While previous research has suggested that TIC may simply be good patient care given that its core components are safety, trust, collaboration, empowerment, and cultural sensitivity [46–48], TIC is unique as it involves understanding patients' experiences through a trauma lens.

The aim of this study was to investigate patients' and physicians' opinions of TIC using a patient-oriented research approach. Overall, TIC was seen as important and already practiced in primary care; however, aspects related to Safety, Trust, and Collaboration were rated higher than Trauma, Empowerment, Peer Support, and Cultural Sensitivity. Informed by these results, patient, family, and physician advisors identified recommendations they thought would be helpful for patients and primary care physicians. These recommendations were in the form of physician training, patient engagement, and system factors.

Physician training may help providers facilitate provision of TIC. TIC physician training specifically may provide physicians with a fundamental knowledge of trauma and its impact on individuals. Further, TIC training may help physicians learn how to foster a sense of safety, trust, empowerment, collaboration, and cultural sensitivity in their patients. Physician training in gender and racial disparities, and marginalized groups could provide physicians with knowledge on how trauma and health differs across groups. Further, this training could help physicians gain insight into how social roles and cultural norms influence patients' perceptions of safety and power dynamics. Emotional skills and self-compassion training could offer additional skills to help physicians understand and respond compassionately to patients' emotions. Patients and physicians reported they would find the aforementioned physician trainings to be helpful and likely to positively impact patient care. Physicians indicated they would likely attend training if CME credit was offered. Healthcare regions/authorities and the Royal College of Physicians and Surgeons of Canada, which is responsible for developing CME programs [67], should consider developing and offering these kinds of training for physicians. To help incentivize physicians to seek this kind of training and to implement TIC in practice, physicians should be adequately compensated for extended appointment times and for providing counselling to their patients.

Information pamphlets and online trauma resource centres may provide patients with the education and support they need to help manage any emotional difficulties. Further, having access to accurate health information could help empower patients to make informed decisions [68, 69] and could promote shared decision-making [68]. Although peer support groups could potentially benefit patients, patients' opinions from this study does not support this recommendation. Provincial healthcare systems should work on improving accessibility of such documents for both physicians and patients through continued education on the resources available.

Online trauma resource centres could provide patients with information to help facilitate understanding of their traumatic experiences and in turn foster a sense of empowerment. Two trauma resource centres already exist in Manitoba [70] and Nova Scotia [71], but there is only one at the national level [72]. While there may also be local centers that provide trauma resources, having centres designated as trauma resource centres may be helpful. This may be a relatively low-cost recommendation to implement to increase patient engagement and

awareness. Similarly, clinical pathways could be developed at a low cost, and could help direct physicians to some of these programs as well as other resources in the community that promote patient wellness.

## Strengths and limitations

While there are studies that investigated physicians' knowledge and attitudes towards TIC [7, 8], there has been little to no investigation of patients' opinions on TIC. The present study attempted to fill a gap in the research by examining perspectives from two different stakeholder groups: patients and primary care physicians.

A significant strength of this study was the active engagement of patient, family, and physician advisors throughout the research process using a patient-oriented research approach. Engaging patients makes research more accountable and transparent, helps provide new insights, and ensures research is relevant for patients [73]. All the recommendations in Phase 2 were directly relevant to patient care as they were formulated in conjunction with patient, family, and physician advisors. Furthermore, integrated knowledge translation (applying principles of knowledge translation to the research process) was applied throughout the duration of the research project as knowledge users (i.e., patients and physicians) were involved [74].

The patient sample in Phase 1 was representative of the Canadian population in regards to the gender, ethnicity, and province of residence. However, Canada's population is approximately 35 million [22] and the perspectives described in this study may not be shared by all in Canada. While efforts were made to recruit primary care physicians from across Canada, only physicians from six provinces and one territory participated. Considering there are 43,500 licensed physicians in family medicine/general practice [75], a sample size of 60 physicians is small and is likely not fully representative of family physicians practicing in Canada.

The TIC survey was created for the purpose of this study as validated measures were not readily accessible at the time this study was developed. While it was piloted on university students and physicians in the community prior to administration, it has not been validated and as such is a limitation.

There could have been potential biases in physicians' responses. Although participants were informed that their responses would be de-identified for analysis, physicians were responding to questions about their own practice and may have responded in a socially desirable way to avoid appearing negative or uncaring. Approximately 850 physicians were individually invited to participate, but only 60 completed Phase 1. Perhaps physicians who participated were interested in the topic, found the topic important, and/or were already practicing some variation of TIC. This may possibly explain the positive responses to the survey items in both Phase 1 and 2.

## Conclusion

Results from Phase 1 showed that both patients and physicians found TIC to be important. While physicians reported they were already delivering aspects of TIC with high frequency, patients indicated they were occasionally receiving them. These findings suggest there is a need for TIC training as patients find TIC important but do not perceive receiving it frequently. Recommendations on physician training, patient engagement, and system factors that affect physicians' ability to provide TIC for their patients were investigated in Phase 2. Results showed these recommendations were acceptable to patients and physicians; however, further research is needed to investigate whether these recommendations improve patient care. Programs that implement these recommendations could be developed and subsequently evaluated to assess improvements in clinical outcomes. TIC is an important part of good patient care,

and can help physicians understand the nuances of how trauma experiences can affect individuals. TIC involves responding to patients in a way that actively avoids re-traumatization, by creating environments that are safe, transparent, supportive, collaborative, empowering, and culturally sensitive [1]. Trauma-informed practices allow physicians and their staff to recognize signs and symptoms of trauma in patients and families, as well as among themselves. Increasing provision of TIC would improve patient care, by promoting healthy relationships between patients and physicians, and more positive healthcare experiences for patients and their families.

## Supporting information

**S1 File. TIC survey.** Patient and physician trauma-informed care survey.
(DOCX)

**S2 File. Phase 2 recommendations.** Patient and physician survey for Phase 2.
(DOCX)

## Acknowledgments

Special thanks to Ms. Tracy Kydd, Dr. John Fryters, Mrs. Hannelore Fryters, Dr. Crystal Ferguson, and Mr. Emiolio Filomeno for their involvement throughout the research process.

## Author Contributions

**Conceptualization:** Seint Kokokyi, Bridget Klest.

**Data curation:** Seint Kokokyi, Bridget Klest.

**Formal analysis:** Seint Kokokyi, Bridget Klest.

**Funding acquisition:** Bridget Klest.

**Investigation:** Seint Kokokyi, Bridget Klest.

**Methodology:** Seint Kokokyi, Bridget Klest.

**Project administration:** Seint Kokokyi.

**Supervision:** Bridget Klest.

**Writing – original draft:** Seint Kokokyi, Hannah Anstey.

**Writing – review & editing:** Seint Kokokyi, Bridget Klest, Hannah Anstey.

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
