## [Decision Letter · Decision Letter 0]

19 Feb 2021

PONE-D-20-27620

A Patient Oriented Research Approach to Assessing Patients’ and Primary Care Physicians’ Opinions on Trauma-Informed Care

PLOS ONE

Dear Dr. Kokokyi,

Thank you for submitting your manuscript to PLOS ONE. After careful consideration, we feel that it has merit but does not fully meet PLOS ONE’s publication criteria as it currently stands. Therefore, we invite you to submit a revised version of the manuscript that addresses the points raised during the review process.

The manuscript has been evaluated by two reviewers, and their comments are available below and attached.

The reviewers have raised a number of concerns that need attention. They request modifications to the reporting in the paper, and also ask about the validation of the questionnaire.

Could you please revise the manuscript to carefully address the concerns raised?

We look forward to receiving your revised manuscript.

Kind regards,

Marianne Clemence

Associate Editor,

PLOS ONE

Journal Requirements:

Furthermore, in the methods section, please address the following:

- a description of any inclusion/exclusion criteria that were applied to participant recruitment.

- a description of how participants were recruited.

3.We note that the grant information you provided in the ‘Funding Information’ and ‘Financial Disclosure’ sections do not match.

Reviewers' comments:

Reviewer's Responses to Questions

**Comments to the Author**

1. Is the manuscript technically sound, and do the data support the conclusions?

Reviewer #1: Yes

Reviewer #2: Yes

2. Has the statistical analysis been performed appropriately and rigorously? 

Reviewer #1: I Don't Know

Reviewer #2: Yes

3. Have the authors made all data underlying the findings in their manuscript fully available?

Reviewer #1: Yes

Reviewer #2: Yes

4. Is the manuscript presented in an intelligible fashion and written in standard English?

Reviewer #1: Yes

Reviewer #2: Yes

5. Review Comments to the Author

Reviewer #1: I am thrilled to see a manuscript with this subject being considered for publication. Trauma-informed care (TIC) is a well-established framework for providing quality clinical care to survivors of various forms of trauma. TIC is rapidly growing in popularity and will soon affect the practice of all subspecialties of medicine. The authors have done an excellent job in conceptualizing and studying trauma-informed interventions that are proposed to impact patient care. The main strength of this article is its inclusion of perspectives from both physicians and patients. The data noted in the attached Tables were helpful and interesting. I have included comments throughout the manuscript with suggestions that I hope will be useful to the authors. One recommendation that I have is to include all introductory information re: trauma, TIC, definitions and epidemiology early on in the manuscript. It would also help to clarify how the authors selected their own key principles of TIC that differ slightly from SAMHSA's. I have also made recommendations for the authors to consider noting other published work on this subject that may be helpful and/or related. References to concepts such as emotional intelligence training in the manuscript can be more specifically tied to TIC (i.e. see Sanctuary Model, 2008). There were several instances throughout the discussion sections where I felt that the voice / insight / explanations of the primary care physician was not as clear as I would have hoped. I wonder whether collaboration with a PCP might be helpful in strengthening the interpretation of the results in the manuscript. Overall, this piece makes an important contribution to the field and would be wonderful to publish following edits as suggested. Thank you for the opportunity to review this paper.

Reviewer #2: Very well written and sound methodologies. The only question I have is regarding the measurement instrument, is it validated. If not, why the research team decided to use this approach--should also note as a limitation if not validated.

6. PLOS authors have the option to publish the peer review history of their article (what does this mean?). If published, this will include your full peer review and any attached files.

Reviewer #1: No

Reviewer #2: No

---

## [Author Response · Author response to Decision Letter 0]

25 Mar 2021

We thank the reviewers for their positive response to our manuscript. The comments provided throughout the manuscript from Reviewer 1 were helpful and we have addressed the reviewer’s suggestions. As suggested, we me included introductory information earlier on in the manuscript and clarified how our key principles of TIC aligned with SAMHSA’s. We reviewed references that were suggested and incorporated them into the discussion as appropriate. We hope that the revised discussion sections are clearer. We have now noted that the measurement instrument has not been validated and as such is a limitation as per Reviewer 2's suggestion. In the Strengths and limitations section, we also added the reason we decided to use this approach.

---

## [Decision Letter · Decision Letter 1]

9 May 2021

PONE-D-20-27620R1

A patient oriented research approach to assessing patients’ and primary care physicians’ opinions on trauma-informed care

PLOS ONE

Dear Dr. Kokokyi,

Thank you for submitting your manuscript to PLOS ONE. After careful consideration, we feel that it has merit but does not fully meet PLOS ONE’s publication criteria as it currently stands. Therefore, we invite you to submit a revised version of the manuscript that addresses the points raised during the review process.

We look forward to receiving your revised manuscript.

Kind regards,

April Joy Joy Damian, PhD, MSc

Academic Editor

PLOS ONE

Journal Requirements:

Reviewers' comments:

Reviewer's Responses to Questions

**Comments to the Author**

1. If the authors have adequately addressed your comments raised in a previous round of review and you feel that this manuscript is now acceptable for publication, you may indicate that here to bypass the “Comments to the Author” section, enter your conflict of interest statement in the “Confidential to Editor” section, and submit your "Accept" recommendation.

Reviewer #1: All comments have been addressed

Reviewer #3: (No Response)

2. Is the manuscript technically sound, and do the data support the conclusions?

Reviewer #1: Yes

Reviewer #3: Yes

3. Has the statistical analysis been performed appropriately and rigorously? 

Reviewer #1: I Don't Know

Reviewer #3: Yes

4. Have the authors made all data underlying the findings in their manuscript fully available?

Reviewer #1: Yes

Reviewer #3: Yes

5. Is the manuscript presented in an intelligible fashion and written in standard English?

Reviewer #1: Yes

Reviewer #3: Yes

6. Review Comments to the Author

Reviewer #1: -I very much appreciate the authors’ hard work in revising this manuscript. It was a completely different experience reading this version compared to the first. Impressive revision indeed! I would be likely to recommend this article to peers in TIC.

-The introduction is much clearer and helpful in defining TIC. The only area for improvement in the Intro is the final paragraph of the introduction. I would define Phase 1 with slightly more detail (ex. “In Phase 1, the research team examined physician and patient opinions of TIC” as well as Phase 2 “…for Phase 2’s examination of physician and patient recommendations for TIC applications”). Otherwise, references to Phase 1 and 2 in the intro lack meaning and leave lingering questions for the reader. Also in this paragraph, “the framework and six core principles” should be further clarified—whose framework? And SAMHSA’s six core principles? It would be best to specify “paraphrased by our team as…”. I would also expand the relevance of each of your principles from “patients” to include everyone in the system, because in a true trauma-informed organization (as defined in Tip 57), SAMHSA describes the principles as relevant for application to clients, family, staff, etc. For Cultural Sensitivity, consider adding a word that goes beyond awareness of biases and includes addressing them.

-The Phase 1 Methods are very well described.

-I love the focus on the IMPACT of trauma in the Discussion section of Phase 1. Physicians may have rated inquiry low b/c of unfamiliarity with trauma screening (PCPs in the U.S. are not trained in this at all)—might be helpful to note that. The sentence “Whereas simply knowing…” doesn’t seem like a complete sentence. There’s a key opportunity in this paragraph to mention “universal precautions” with TIC—it may not actually be relevant to know your pt’s trauma hx, if you apply these principles universally.

-In the Phase 2 section, 1st paragraph, would clarify “…patients did not perceive the __importance of?__ receiving TIC at the same level…”

-In the Phase 2 Results 3rd paragraph, would add an apostrophe for “…lower than physicians’…”

-In the final paragraph of the Phase 2 Results section, it would make more sense to switch these 2 sentences, as the phrase “clinical pathway” comes as a surprise to readers (hasn’t yet been mentioned). It would also be helpful to define what you mean by clinical pathway and what role it played in your study.

-In the Conclusion of the paper, I might adjust the 3rd sentence to more accurate reflect your results. Consider something to the effect of “…as pts find TIC important but do not perceive receiving it frequently.” I would strongly consider ending the conclusion on a positive note, rather than closing with a statement that includes the phrase “cost-effective”. Why do you think TIC is important? Do you think it’s possible to implement these strategies you’ve identified? What do you think would be the ultimate impact on physicians and patients, or healthcare on the whole?

Reviewer #3: This is an important investigation - there must be an understanding of how both the patient and the provider view TIC in order to increase the probability that the two will connect in a way that enhances the healing experience of the patient and gives instruction to the provider regarding how better to provide care. This study shows areas of concordance and discord between the two groups, which is a good starting point. The discussion of the reasons behind the answers endorsed be each group is necessarily speculative 9in the absence of focus groups or another method to get to the thought processes of providers and patients leading to the results. The discussion suggests that social determinants of health and historical institutionalized racism can be viewed through a trauma lens and this is likely a fruitful area for future work.

7. PLOS authors have the option to publish the peer review history of their article (what does this mean?). If published, this will include your full peer review and any attached files.

Reviewer #1: No

Reviewer #3: No

---

## [Author Response · Author response to Decision Letter 1]

12 Jun 2021

We thank both reviewers for their helpful and supportive feedback. We addressed each of the comments from Reviewer #1 in the respective sections of the manuscript and provided a version with track changes to highlight the revisions made. As suggested, we included a statement about paraphrasing SAMHSA’s TIC principles, we delineated the role of clinical pathways, and we expanded on the conclusion. In regards to Reviewer #3's comments, future research will be needed to investigate how the recommendations proposed will improve patient care. We hope this revised draft is clearer.

---

## [Editor Report · Decision Letter 2]

24 Jun 2021

A patient oriented research approach to assessing patients’ and primary care physicians’ opinions on trauma-informed care

PONE-D-20-27620R2

Dear Dr. Kokokyi,

We’re pleased to inform you that your manuscript has been judged scientifically suitable for publication and will be formally accepted for publication once it meets all outstanding technical requirements.

Kind regards,

April Joy Joy Damian, PhD, MSc

Guest Editor

PLOS ONE
---

## [Editor Report · Acceptance letter]

30 Jun 2021

PONE-D-20-27620R2 

A patient-oriented research approach to assessing patients’ and primary care physicians’ opinions on trauma-informed care 

Dear Dr. Kokokyi:

I'm pleased to inform you that your manuscript has been deemed suitable for publication in PLOS ONE. Congratulations! Your manuscript is now with our production department. 

Kind regards, 

on behalf of

Dr. April Joy Joy Damian 

Guest Editor

PLOS ONE